# Determination of Greenhouse Gas Concentrations from the 16U CubeSat Spacecraft Using Fourier Transform Infrared Spectroscopy

**DOI:** 10.3390/s23156794

**Published:** 2023-07-29

**Authors:** Vera Mayorova, Andrey Morozov, Iliya Golyak, Igor Golyak, Nikita Lazarev, Valeriia Melnikova, Dmitry Rachkin, Victor Svirin, Stepan Tenenbaum, Ivan Vintaykin, Dmitriy Anfimov, Igor Fufurin

**Affiliations:** 1Special Machinery Department, Bauman Moscow State Technical University, 105005 Moscow, Russia; mayorova@bmstu.ru (V.M.); lnd18m350@student.bmstu.ru (N.L.); melnikovabg@bmstu.ru (V.M.); rachkin@bmstu.ru (D.R.); tenenbaum@bmstu.ru (S.T.); 2Physics Department, Bauman Moscow State Technical University, 105005 Moscow, Russia; amor59@mail.ru (A.M.); iliyagol@mail.ru (I.G.); igorgolyak@yandex.ru (I.G.); dd.cpf@yandex.ru (V.S.); vintaikin_ivan@mail.ru (I.V.); diman_anfimov@mail.ru (D.A.)

**Keywords:** CubeSat, remote sensing, greenhouse gas, emissions, air pollution monitoring, FTIR spectroscopy

## Abstract

Greenhouse gases absorb the Earth’s thermal radiation and partially return it to the Earth’s surface. When accumulated in the atmosphere, greenhouse gases lead to an increase in the average global air temperature and, as a result, climate change. In this paper, an approach to measuring CO${}_{2}$ and CH${}_{4}$ concentrations using Fourier transform infrared spectroscopy (FTIR) is proposed. An FTIR spectrometer mockup, operating in the wavelength range from 1.0 to 1.7 μm with a spectral resolution of 10 cm${}^{-1}$, is described. The results of CO${}_{2}$ and CH${}_{4}$ observations throughout a day in urban conditions are presented. A low-resolution FTIR spectrometer for the 16U CubeSat spacecraft is described. The FTIR spectrometer has a 2.0–2.4 μm spectral range for CO${}_{2}$ and CH${}_{4}$ bands, a 0.75–0.80 μm range for reference O${}_{2}$ bands, an input field of view of 10${}^{-2}$ rad and a spectral resolution of 2 cm${}^{-1}$. The capabilities of the 16U CubeSat spacecraft for remote sensing of greenhouse gas emissions using a developed FTIR spectrometer are discussed. The design of a 16U CubeSat spacecraft equipped with a compact, low-resolution FTIR spectrometer is presented.

## 1. Introduction

There are many point sources that produce anthropogenic greenhouse gas emissions, including coal mines, landfills, sewage treatment facilities and animal husbandry. Although current ground-based detection methods can detect gases at low thresholds, their application time and coverage area are constrained. Rapid and precise identification of greenhouse gas emissions can be achieved through satellite monitoring. Deploying a constellation of satellites can remove restrictions on the observation time.

Hydrometeorological spacecraft, such as Meteor-M [1], Meteorological Operational Satellite (MetOp) [2], Terra/Aqua and Sentinel-5P (TROPOMI) [3,4], which are equipped with infrared spectrometric and multispectral/hyperspectral instruments, have been used successfully to solve climate monitoring tasks. Such equipment allows estimating the atmosphere’s chemical composition (spatial resolution of 10–100 km, viewing bandwidth up to 2500 km). The Methane Remote Sensing Lidar Mission (MERLIN) [5] equipped with a lidar instrument is planned for 2027 and has been designed for arctic and nighttime observations. There are also specialized satellites, such as Greenhouse Gases Observing Satellite (GOSAT) [6,7], Orbiting Carbon Observatory 2 (OCO-2) and OCO-3 [8], equipped with infrared spectrometers for more accurate localization when determining the content of greenhouse gases in the Earth’s atmosphere than the equipment of hydrometeorological devices (spatial resolution of 1–10 km, viewing bandwidth up to 100 km). The spacecraft Landsat-8 [9], WorldView-3 [10] and Sentinel-2 [11] have achieved very high spatial resolutions (up to 30 m) with global coverage. These devices are distinguished by their high-quality technical characteristics, large productivity and large amount of measured data. Environmental Mapping and Analysis Program (EnMAP) [12] and Precursore IperSpettrale della Missione Applicativa (PRISMA) [13] satellites have medium sensitivities, but extensive coverage (about 4 days). In 2016, a project called Greenhouse Gas Satellite (GHGSat) (Canada) [14,15] was developed for a small satellite constellation. The Thermal and Near-Infrared Sensor for Carbon Observation Fourier Transform Spectrometer (TANSO-FTS) onboard the Greenhouse Gases Observing Satellite is used for remote sensing of atmospheric methane emissions [16]. Nowadays, the Meteor-M 2 [17] spacecraft is used for climate monitoring, and four more of them are planned for launch by 2025 in Russia.

Spectrometers placed on board spacecraft are a marker of the spacecraft’s capabilities for the remote measurement of greenhouse gases. The OCO-2 [18] instrument uses a near infrared (NIR)/short-wave infrared (SWIR) grating spectrometer, with three bands: 0.76 μm (O${}_{2}$), 1.61 and 2.06 μm (CO${}_{2}$). Its spatial resolutions are 1.29 km (cross-track) and 2.25 km (along track). It orbits at an altitude of 705 km, flying in a polar, sun-synchronous formation. The instrument covers a 1.29 by 2.25 km footprint at Nadir and is capable of acquiring eight cross-track footprints, creating a swath width of 10.3 km. The spacecraft launch mass is 449 kg. The Thermal and Near-Infrared Sensor for Carbon Observation Fourier-Transform Spectrometer (TANSO-FTS) [16] onboard the GOSAT has a four-band interferometer (three in the NIR/SWIR and one in the thermal infrared) and is used for CH${}_{4}$, CO${}_{2}$, H${}_{2}$O, O${}_{2}$ and O${}_{3}$ measurements. The field of view (FOV) is 15.8 mrad (10.5 km on the ground at Nadir), and the measurement speed is 1.1, 2, 4 s per interferogram. The TANSO-FTS spectrometer’s mass is 329 kg and its spatial resolution is 10.5 km [19]. GHTSat is an imaging spectrometer [15] based on the Fabry–Perot spectrometer designed for methane monitoring. The described method has limited sensitivity and was designed for a specific spectral line and, as a consequence, for a specific gas. The data obtained from GHGSat devices have already demonstrated their significance and relevance in terms of searching for point sources of pollution. The GHGSat-D satellite’s mass is 15 kg, its spatial resolution is 50×50 m${}^{2}$ and it has a 12×12 km${}^{2}$ coverage [20]. The IKFS-2 FTIR spectrometer operating onboard the Meteor-M 2 satellite is described in [21]; it allows measurements of the gas composition of the atmosphere with a maximum bandwidth of up to 2500 km (with a point step of 100 km). The IKFS-2 FTIR spectrometer’s instantaneous field of view (IFOV) is 40 mrad (35 km), its spectral range is 660–2000 cm${}^{-1}$ and its mass is about 50 kg [22]. The GaoFen-5 satellite-II (GMI-II) uses spatial heterodyne spectroscopy (SHS) for quantitative monitoring of atmospheric greenhouse gases [23]. GaoFen-5 was developed by the China Aerospace Science and Technology Corporation, and has a liftoff mass of 2700 kg and a design life of 8 years [24]. GMI-II uses four IR bands for O${}_{2}$, CO${}_{2}$ and CH${}_{4}$ in the visible and NIR spectral regions. The instantaneous spatial resolution is 10.3 km and the spectral resolution is 0.6 cm${}^{-1}$ for the O${}_{2}$ band and 0.27 cm${}^{-1}$ for the CO${}_{2}$ and CH${}_{4}$ bands.

The devices for measuring greenhouse gases discussed above have properties that make them useful for resolving a variety of issues in remote sensing of the Earth’s properties. However, the creation of instruments for spacecraft in the CubeSat design is currently a critical endeavor. Due to its size and weight, most sensing equipment cannot be fitted in a CubeSat spacecraft. Even a low-resolution FTIR spectrometer such as the Bruker EM27/SUN [25] cannot be placed in the CubeSat.

MeznSat-A is a 3U CubeSat for monitoring greenhouse gases using short-wave infrared (SWIR) spectroscopy. The spacecraft is equipped with an Argus 2000 spectrometer. The spectral range of the grating spectrometer is 1.0 to 1.65 μm, and its spectral resolution is 6 nm across 100 spectral channels. In [26], a ground-based prototype of a multichannel laser heterodyne spectroradiometer for measuring CO${}_{2}$ and CH${}_{4}$ molecular absorption with an ultra-high spectral resolution of 0.0013 cm${}^{-1}$ for a 2U CubeSat was developed. Fellgett’s advantage [27] can produce a relative improvement in the signal-to-noise ratio (SNR) in FTIR spectrometers. Therefore, creating a compact FTIR spectrometer for the CubeSat platform is a crucial task for greenhouse gas remote sensing.

Today, many CO${}_{2}$ measurements are based on NDIR (non-dispersive infrared) spectroscopy [28,29]. Devices of this type are simple, compact and quite cheap, but their spectral resolution is quite low (the filter bandwidth is more than 100 nm) [30]. FTIR spectrometers allow the recording of spectra in a wide range with a high resolution. This makes it possible to measure several greenhouse gas concentrations [31] simultaneously. A comparison of FTIR and NDIR measurements has shown a smaller variance in FTIR measurements [32].

This paper is devoted to the analysis of using infrared spectroscopy for remote measurements of greenhouse gas concentrations and the development of a compact, low-resolution FTIR spectrometer for the CubeSat spacecraft. We focus on CubeSat satellites because they are the cheapest to launch into space as an additional payload. Monitoring should be carried out by CubeSat spacecraft to solve the task of remote sensing in the field of greenhouse gas emissions.

## 2. Materials and Methods

### 2.1. Spacecraft Design

The external and internal layout of the developed small spacecraft follows the logic of the CubeSat standard [33]; the spacecraft has the shape of a parallelepiped (Figure 1). The volume of the spacecraft is 16U and the cross-section of the spacecraft corresponds to the 12U CubeSat standard size of 226×226 mm. Four parallel ribs form the guide rails for the spacecraft to exit the transport and launch pads. The satellite is divided into two parts: the Fourier spectrometer unit and the service system unit. Each of the compartments is installed on two detachable frames with special connecting frames. The main technical characteristics of the device are given in Table 1.

### 2.2. FTIR Spectrometer Unit

An FTIR spectroradiometer unit is integrated into the design of the small spacecraft developed for greenhouse gas remote sensing. Namely, to determine the oxygen and carbon dioxide concentrations in the air along the line of sight of the device, it measures the reflected sun radiation which passes the air column twice (Figure 2). Ground-based prototypes are described in papers [34,35,36]. Substance concentrations are determined by the depth of spectral lines according to the Beer–Lambert law [35,37,38].

Carbon dioxide molecules have fundamental absorption bands around 1.61 μm and 2.05 μm, which can be measured by analyzing reflected IR radiation from the Earth’s surface. The concentration of molecular oxygen O${}_{2}$ is constant, well known and uniformly distributed throughout the atmosphere [39]. The use of an additional channel on the fundamental oxygen absorption line (1.27 μm or 0.765 μm) [40] creates a reference channel to calibrate the accuracy of greenhouse gas concentration measurements by the developed system [41].

The device is based on a Michelson interferometer (Figure 3). Satellite and scientific instruments provide the target image and spectral information about the content of the target gases; O${}_{2}$ is the first channel and CO${}_{2}$ is the second channel. A distinctive feature compared to other missions is the combination of high spatial resolution and low spectral resolution in a compact volume.

The operating principle of the FTIR spectrometer is as follows: Solar radiation reflected from the Earth’s surface passes through the atmosphere and is collected by the telescopic system (Figure 3, positions 1 and 2) of the spectrometer. To reduce the dimensions and optimize the layout, the optical axis of the spectrometer system is deflected by mirror 3 (Figure 3, position 3). The radiation, collected by the telescope by falling on mirror 3, is reflected from it and enters the interferometer. The interferometer includes a beam splitter (Figure 3, position 4) consisting of a beam splitter plate, a compensator plate made of ZnSe and two corner reflectors (Figure 3, positions 5 and 6). We use corner reflectors in the receiving channel of the spectrometer to ensure the stability and reliability of its operation (Figure 3, positions 5 and 6) [36,42] as mirrors with an aperture of 25 mm and a deviation of 1 angular second. Due to the longitudinal movement of the corner reflector ULPR-10 (PLX) (Figure 3, position 6) relative to the fixed corner reflector (Figure 3, position 5), a phase difference occurs and an interferogram is formed at the output. We mounted the movable reflector on a spring parallelogram, which provides a stroke of 4 mm in one direction. The FTIR raw signal (interferogram) has about $3\times 10^{4}$ points, which corresponds to a spectral resolution of 2 cm${}^{-1}$ [43]. The radiation emerging from the interferometer falls on a splitter plate (dichroic filter, Figure 3, position 7) installed at an angle of 45 degrees. A dichroic filter is designed for spectral separation of the recorded radiation. The splitter reflects 98% of the radiation in wavelengths from 0.7 μm to 1.2 μm and transmits 95% in wavelengths from 1.2 μm to 2.4 μm. The transmitted radiation passes through a bandpass filter (Figure 3, position 8) and is collected by the focusing lens (Figure 3 position 9) on the photodetector (Figure 3 position 10). A signal channel is used for the registration of carbon dioxide and methane absorption. The signal is recorded by an InGaAs photodetector IG26X1000T9 (Laser components) (Figure 3 position 10) with an active area of 1 mm and a detectivity of $D^{*}∼2\times 10^{12}\;\mathrm{cm}\times {\mathrm{Hz}}^{1/2}$/W. A bandpass filter (Figure 3 position 8) installed in front of the photodetector (Figure 3 position 9) passes radiation in the spectral range of 2.0–2.3 μm. The radiation reflected from the filter (Figure 3 position 7) passes through the bandpass filter (Figure 3 position 11) and is focused by the lens (Figure 3 position 12) on the photodetector PIN-040A (OSI Optoelectronics) (Figure 3 position 13). This channel is a reference channel used for registration of oxygen absorption bands. O${}_{2}$ absorption band registration is carried out by a Si photodetector (Figure 3, position 13) with a linear size of w = 1 mm. The noise equivalent power (NEP) value for the Si detector is $6.2\times 10^{-15}$ W/Hz${}^{1/2}$. To cut off visible radiation, a band-pass filter (Figure 3, position 11) is also installed in front of the receiver for the spectral range of 0.75–0.80 μm.

The bandpass filter in the signal channel (Figure 3, position 8) is optional and can be excluded from the optical subsystem. Exclusion of this filter would allow expanding the operating spectral range to the limits of the sensitivity of the InGaAs photodetector (1.0–2.5 μm). The extended range would allow simultaneous recording of spectral lines of CO${}_{2}$ (1.6 and 2.05 μm), CH${}_{4}$ (1.66 and 2.36 μm), O${}_{2}$ (0.76 and 1.27 μm) and CO (2.34 μm).

In order to stabilize the speed of the mirror and determine the moments to read the interferogram, a reference channel with a fourfold difference in the optical path is used. The reference channel includes a dihedron (Figure 3, position 17) and a flat mirror (Figure 3, position 18) [42]. A He–Ne laser with a wavelength of 632 nm is used as a reference source. To take into account the instrumental function of the spectrometer [43,44] and adjust the recorded IR absorption spectra, a reference radiation source (small filament lamp MH 1.25–0.25) (Figure 3, position 14) is installed.

The described FTIR spectrometer can be used for trace gas analysis [44] and remote sensing [34]. Mathematical methods of spectral analysis are described in [45,46]. The main technical characteristics of the compact, low-resolution spacecraft-based spectrometer are given in Table 2.

### 2.3. Positioning an Object on the Ground

With a focal length f = 0.1 m (Figure 4), the FOV will be γ = 10${}^{-2}$ rad. The instantaneous size of the ground object is about 5.75 km for an orbit altitude of 575 km. The resulting area size corresponds to a GOSAT TANSO-FTS observation footprint of 10.5 km [47]. The Carbon Dioxide Monitoring Mission (MicroCarb) has a Nadir pixel size of 4.5×9 km${}^{2}$ and The Geostationary Carbon Cycle Observatory (GeoCarb) has a Nadir pixel size of 10×10 km${}^{2}$ [20].

The positioning accuracy provided by the satellite’s orientation system is 0.1${}^{\circ }$, which corresponds to 1 km on the Earth’s surface (the orientation error is 17%). The average motion of the satellite and its subsatellite point are:(1)$$n=\sqrt{\frac{\mu {}_{E}}{{\left(R{}_{E}+h\right)}^{3}}}=0.0625\;\deg/s,$$
where μ${}_{E}$ is Earth’s standard gravitational parameter, $R_{E}$ is the Earth’s radius and orbital altitude *h* = 575 km. Then, the velocity of the subsatellite point will be $V_{E}$ = $R_{E}$ × n = 6.95 km/s. We assume that the FTIR spectrometer interferogram measuring rate is 10 Hz and the measurement time t is 0.1 s. The satellite orientation error will be s${}_{E}$ = V${}_{E}$ × t = 695 m (12%). The resulting error is adequately correlated with the size of the object on the ground and allows scanning of the Earth’s surface based on the relative motion of the satellite in orbit.

For the described FTIR spectrometer, the SNR in the recorded IR spectrum was estimated for a registration time of 0.1 s and in an orbit of 575 km. The calculation took into account the parameters of the spectrometer, such as the NEP of the detector, entrance aperture and transmission of the optical system, which is equal to 0.3. The estimated SNR for spectral band 2.0–2.4 μm is about 450.

### 2.4. Service System Unit

The service system boards were installed in the service system unit on four racks. The preliminary sequence of installing the boards in each rack is shown in Figure 5. The racks are mounted on a large control board. The service systems were developed taking these factors into account during the small spacecraft flight tests [48].

Due to the requirements for the orientation of the photodetector at Nadir, the satellite has an active three-axis altitude control system consisting of flywheels [49] with unloading by magnetic coils. Two magnetic coils are located on each of the external panels on the X and Y axes and four are located on the Z axis of the control board. To determine the orientation of the spacecraft, data from the star sensor, solar sensor, accelerometer and magnetometer are used.

The power supply system provides 80 W of power and a voltage of 12 V to the equipment. The batteries are LiFePO4 elements of the ANR26650 type. These elements are characterized by a large resource and a high current output. Solar panels are built on the base of AsGa solar cells with an efficiency of 28.8 % [50].

For transferring the target information to the ground control center, radio communication systems in the UHF, S and X bands are used.

To change the satellite’s orbit, three electric propulsion engines are used [51,52]. It is proposed to use Teflon as a working medium, since it has a high density (2.21 g/cm${}^{3}$) and a low after-evaporation, a factor that reduces the characteristics of an ablative pulsed plasma engine.

## 3. Results and Discussion

A spacecraft mockup was created (Figure 6a). External panels (including the installation of solar cells and antennas), frames, a mockup of the star sensor, a block of service systems (including the boards of systems and devices), a propulsion system and a camera were manufactured.

The satellite’s weight is about 23 kg, its dimensions are 23 × 23 × 46 cm and 43% of its mass falls on the payload. Designed with proliferation and manufacturability in mind, the satellite is optimized for greenhouse gas monitoring tasks by using readily available parts (e.g., industrial electronic components) and inexpensive production conditions.

An FTIR spectrometer mockup (Figure 6b) for ground-based tests of greenhouse gas concentration measurements was created. The technical characteristics of the FTIR mockup are given in Table 3.

Experimental measurements were taken in September 2022 near the Baumanskaya district (Moscow, Russia). Solar radiation passes through the air masses of the atmosphere and is reflected off both inhomogeneities of the atmosphere and off topographic objects. The mockup registered reflected solar radiation. The line of sight direction did not change during the experiment. The overall time of measurements was 5 h. The ambient temperature was 12 ${}^{\circ }$C and did not change significantly during the experiment. The humidity was 83%. The signal was recorded in cloudless weather.

Figure 7 shows the scheme (Figure 7a) of solar reflection measurements in urban conditions and the appearance of the mockup (Figure 7b). The mockup of the spectrometer (Figure 7b, position 1) was installed outdoors on a tripod (Figure 7b, position 2), which allowed it to be oriented in the desired direction. The mockup was powered by two Rigol DP832 laboratory power sources (Figure 7b, position 3), which were located next to it. The recorded signal from spectrometer was transmitted to a laptop (Figure 7b, position 4) for processing. The FTIR mockup was oriented towards the open terrace at about 20 degrees to the horizon.

The OCO retrieval algorithm [53] uses the fitting of the O${}_{2}$ A-band at 0.76 μm and the CO${}_{2}$ bands at 1.61 μm and 2.06 μm, and it can be applied to many different retrieval problems. We used a type of OCO algorithm for the absorption bands of O${}_{2}$ at a wave number of 7880 cm${}^{-1}$ that is appropriate for the CO${}_{2}$ and CH${}_{4}$ bands at 6250–6350 cm${}^{-1}$ and 6024 cm${}^{-1}$, respectively.

The spectral transmittance of the atmosphere obtained with an FTIR spectrometer is shown in Figure 8. The spectrum was obtained by averaging 15 interferograms with a total recording time of 1 min. The experimental spectrum includes oxygen absorption lines of O${}_{2}$ at the wave number of 7880 cm${}^{-1}$, CO${}_{2}$ at the wave numbers of 6250 cm${}^{-1}$ and 6350 cm${}^{-1}$ and CH${}_{4}$ at 6024 cm${}^{-1}$. For a single measurement, the SNR in the spectrum is equal to 1220. The positions of the absorption lines for O${}_{2}$, CO${}_{2}$ and CH${}_{4}$ correspond to the ones given in the HITRAN spectral database [54], see Figure 9. We used open source software such as Python, NumPy, SciPy and Matplotlib for data processing and visualization.

Graphs of CO${}_{2}$ and CH${}_{4}$ concentrations are shown in Figure 10. The shape of the graph corresponds with traffic jams in the observation area.

Measurements of greenhouse gases were obtained from the transmittance spectra in Figure 8 by constructing a baseline on the characteristic spectral bands for each substance (6100–6400 cm${}^{-1}$ for CO${}_{2}$, 7766–8009 cm${}^{-1}$ for O${}_{2}$ and 5980–6010 cm${}^{-1}$ for CH${}_{4}$). The baseline was calculated as a linear function with three points at the boundaries of each interval. Integral concentrations (ppm × m) of CO${}_{2}$ and CH${}_{4}$ were calculated from the depth of the absorption lines in the recorded spectra. The volumetric concentrations (ppm) of CO${}_{2}$ and CH${}_{4}$ were calculated from their integral concentrations. The obtained values were smoothed by a sinc window function with a half-width of 50 points. The obtained values of CH${}_{4}$ concentrations correspond to the data from MosEcoMonitoring measuring stations [55] and CO${}_{2}$ concentrations correspond to the data in [56]. The variations in CO${}_{2}$ concentration during the day correspond to the data from near-surface air measurements using a Fourier transform spectrometer [57,58].

The presented results experimentally demonstrate the possibility of measuring greenhouse gas concentrations by reflected solar radiation using an IR Fourier spectrometer. The developed compact, low-resolution FTIR spectrometer is designed to be placed in CubeSat spacecraft for remote measurement of greenhouse gas concentrations.

The functional parts of the CubeSat FTIR spectrometer are the same as for the ground-based mockup. The differences are in the mechanical design, for example, the use of special materials with an extended temperature range. To limit the radiation flux, a motorized diaphragm can be installed in the optical system. A diaphragm allows us to change the size of the radiation flux in the range from 1 mm to 27 mm. To ensure functioning in outer space, special heat sinks from electronic circuit boards can be used, special vacuum lubrication of moving parts can be provided and a radiation-resistant element base can be used. The CubeSat FTIR spectrometer will be tested in conditions close to those in outer space.

For the CubeSat FTIR spectrometer, we assume a detection limit of about 1% of the nominal values of CO${}_{2}$ and CH${}_{4}$ concentrations in the atmosphere. The precision and accuracy of gas detection and concentration evaluation are determined by the spectral resolution of the device at the level of 2 cm${}^{-1}$. In the considered measurement range, the main atmospheric components do not have strong absorption lines according to HITRAN [54] and GEISA databases and SPECTRA Spectroscopy Tools [59].

## 4. Conclusions

An FTIR spectrometer mockup for recording IR absorption spectra in the wavelength range from 1.0 to 1.7 μm with a spectral resolution of δν=10 cm${}^{-1}$ is described. The CO${}_{2}$ absorption line with a central wavelength λ = 1.60 μm, the absorption line of CH${}_{4}$ with λ = 1.66 μm and the absorption line of O${}_{2}$ with λ = 1.27 μm were recorded from the reflected solar radiation. The atmospheric transmittance was monitored throughout the day in urban conditions. The obtained data allowed monitoring of the integral and volumetric concentrations of CO${}_{2}$ and CH${}_{4}$. It was shown that the time dependence of CO${}_{2}$ and CH${}_{4}$ concentrations well reflects the degree of traffic congestion on that day.

We developed an optical scheme (Figure 3) and design (Table 2) of a compact low-resolution FTIR spectrometer. Small spacecraft like CubeSat 16U can use the spectrometer for remote sensing of multiple greenhouse gases. The instantaneous size of the ground object is about 5.75 km at an orbit altitude of 575 km, the spectral resolution is 2 cm${}^{-1}$, the spectral ranges are 0.75–0.80 μm and 2.0–2.4 μm and the mass is less than 10 kg. The estimated SNR for spectral band at 2.0–2.4 μm is about 450. The detection limit for the described spectrometer is about 1% of the nominal values of CO${}_{2}$ and CH${}_{4}$ concentrations in the atmosphere. The precision and accuracy of gas detection and concentration evaluation are determined by the spectral resolution at a level of 2 cm${}^{-1}$. To calculate CO${}_{2}$ and CH${}_{4}$ concentrations, we use normalization with a 0.75–0.8 μm O${}_{2}$ A-band.

The FTIR spectrometer allows for simultaneous concentration measurements of several greenhouse gases and is designed for CubeSat spacecraft.

## Figures and Tables

**Figure 1 sensors-23-06794-f001:**
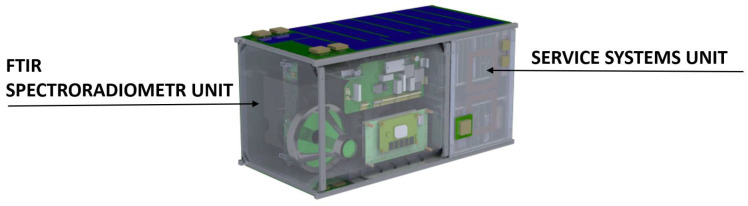
Spacecraft internal design.

**Figure 2 sensors-23-06794-f002:**
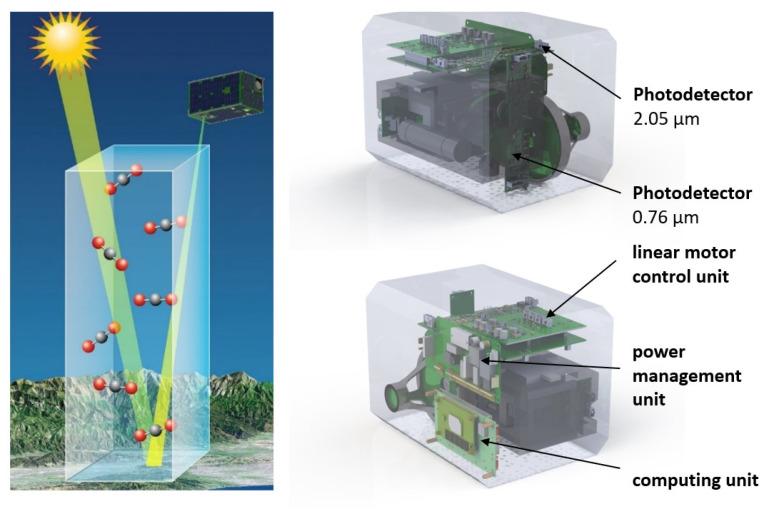
FTIR spectrometer principle of operation and unit design.

**Figure 3 sensors-23-06794-f003:**
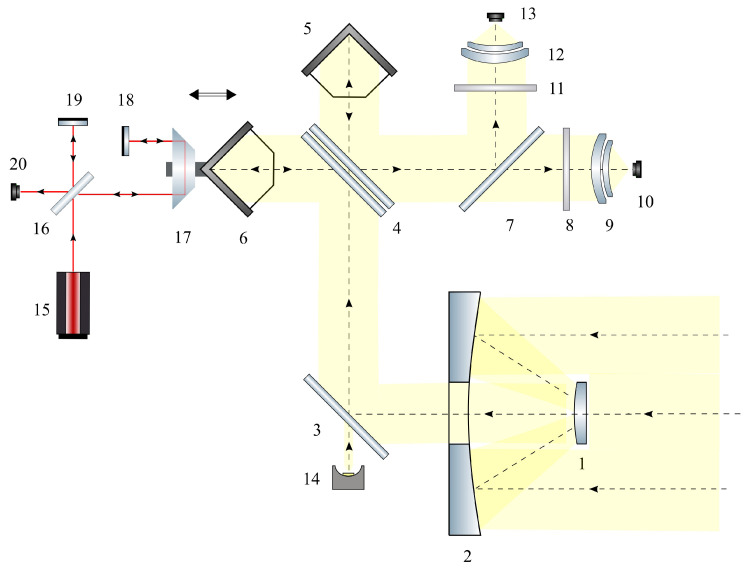
The compact low-resolution FTIR spectrometer optical scheme: 1, 2—mirror lens; 3—mirror; 4—ZnSe beam splitter; 5, 6—retroreflectors; 7—dichroic filter; 8—band-pass filter 2.0–2.4 μm; 9, 12—focusing lens; 10—photodetector 2.0–2.4 μm; 11—band-pass filter for a wavelength of 0.75–0.80 μm; 13—photodetector 0.75–0.80 μm; 14—reference radiation source; 15—632 nm laser; 16—reference channel beam splitter; 17—dihedron; 18, 19—flat mirrors; 20—reference channel photodetector.

**Figure 4 sensors-23-06794-f004:**
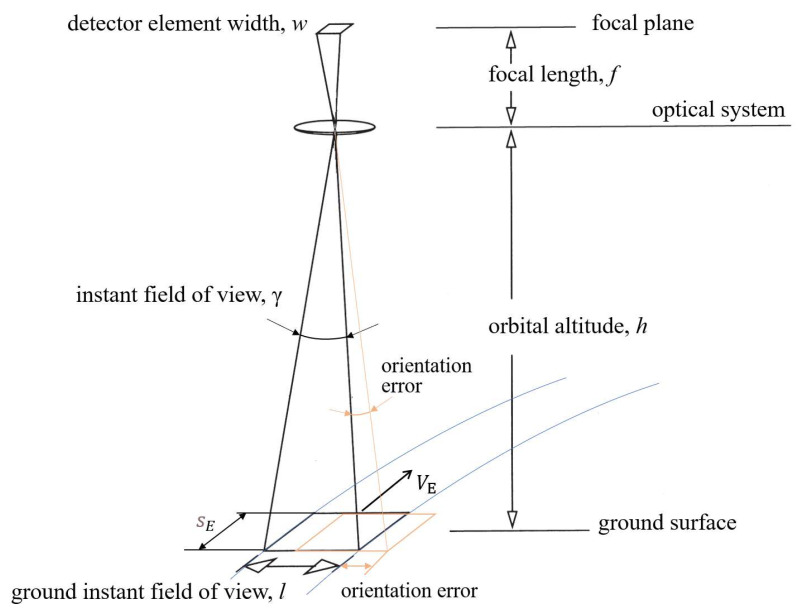
The basic arrangement of the detector element and the viewing area.

**Figure 5 sensors-23-06794-f005:**
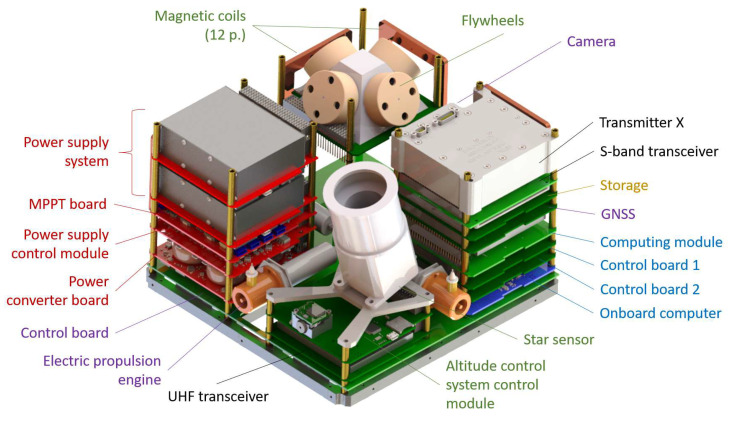
Racks of boards with service systems.

**Figure 6 sensors-23-06794-f006:**
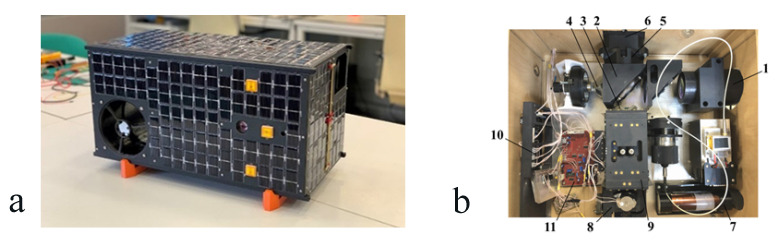
The mockup design of a spacecraft (**a**) and FTIR spectrometer design (**b**): 1—input lens; 2—beam splitter; 3, 4—corner reflectors; 5—focusing lens; 6—signal IR photodetector IG17X2000T9 (Laser Components); 7—632 nm laser; 8—reference photodetector PIN-040A (OSI Optoelectronics); 9—parallelogram; 10—power supply board; 11—linear drive control board.

**Figure 7 sensors-23-06794-f007:**
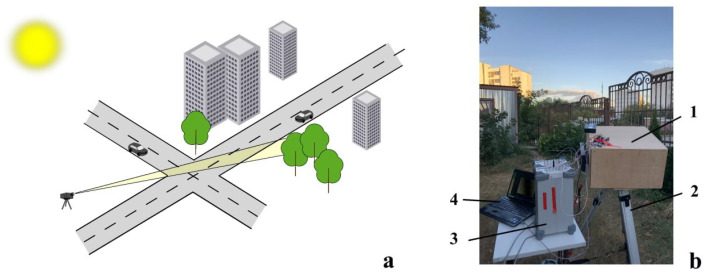
The scheme of solar reflection measurements in urban conditions: 1—spectrometer; 2—tripod; 3—Rigol DP832 laboratory power sources; 4—laptop.

**Figure 8 sensors-23-06794-f008:**
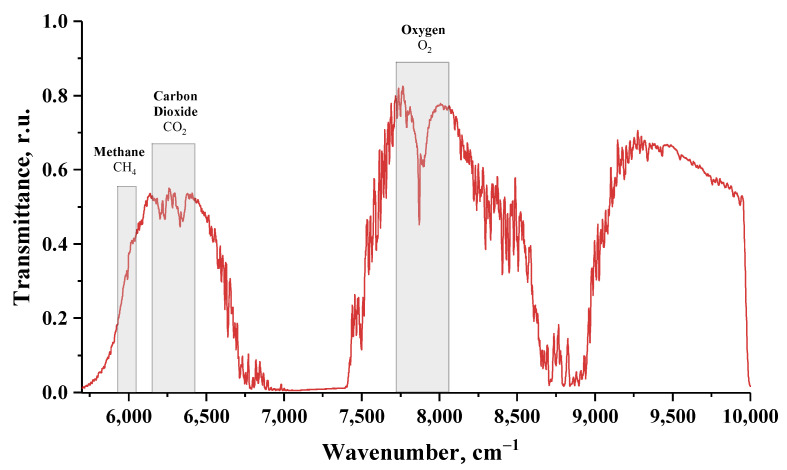
The atmosphere transmittance spectrum measured by the FTIR spectrometer mockup. The spectral bands for O${}_{2}$ at 7880 cm${}^{-1}$, CO${}_{2}$ at 6250 and 6350 cm${}^{-1}$ and CH${}_{4}$ at 6024 cm${}^{-1}$ are shown.

**Figure 9 sensors-23-06794-f009:**
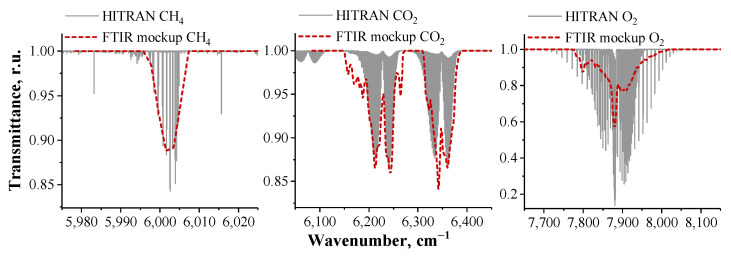
The HITRAN and measured transmittance spectra for O${}_{2}$, CO${}_{2}$ and CH${}_{4}$.

**Figure 10 sensors-23-06794-f010:**
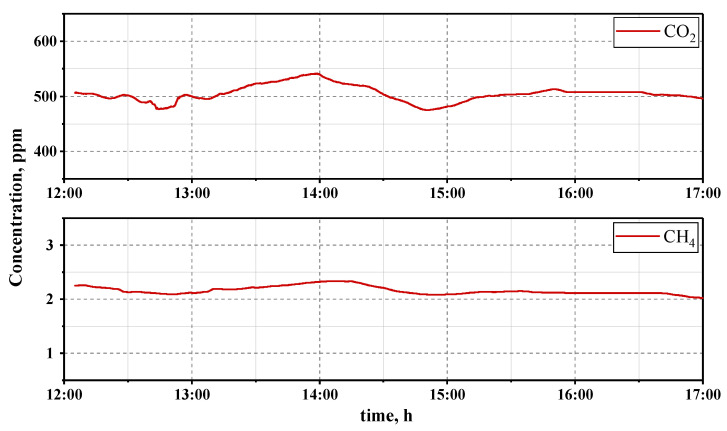
Registered carbon dioxide and methane concentrations throughout a day.

**Table 1 sensors-23-06794-t001:** Spacecraft technical characteristics.

Parameter	Unit	Value
Lifetime ${}^{1}$	years	no less than 3
Dimensions	mm	226×226×454 (16U CubeSat)
Weight	kg	23
Orbit parameters	km	from 500 to 600 km, SSO
Daily power (average orbital)	W	10
Altitude control system		Triaxial: flywheels with unloading by magnetic coils
Orientation error (3σ) on all axes	deg	no more than 0.1
Stabilization error (3σ) on all axes	deg/s	no more than 0.01
Propulsion system		electric ablative pulse
Payload		FTIR spectrometer

${}^{1}$ The lifetime of the spacecraft is limited by the insufficient electrical power supply system parameters.

**Table 2 sensors-23-06794-t002:** The compact low-resolution FTIR spectrometer characteristics.

Parameter	Unit	Value
Spectral range:		
O${}_{2}$	μm	0.75–0.80
CO${}_{2}$ and CH${}_{4}$	μm	2.0–2.4
Spectral resolution	cm${}^{-1}$	2
FOV	rad	10${}^{-2}$
Entrance aperture	mm	100
Power consumption	W	up to 100
Dimensions	mm	268×208×216
Weight	kg	no more than 10

**Table 3 sensors-23-06794-t003:** The FTIR spectrometer mockup technical characteristics.

Parameter	Unit	Value
Spectral range	μm	1.0–1.7
Spectral resolution	cm${}^{-1}$	10
FOV	deg	4
Entrance aperture	mm	100
IR photodetector material		InGaAs
IR active area size	mm	2
IR Photodetector NEP	W/Hz${}^{1/2}$	$1.7\times 10^{-15}$

## Data Availability

The data presented in this study are available on request from the corresponding author.

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
