# Peer review of "Determination of Greenhouse Gas Concentrations from the 16U CubeSat Spacecraft Using Fourier Transform Infrared Spectroscopy"

_sensors, 2023, doi:10.3390/s23156794_

Round 1

Reviewer 1 Report

- The authors should eliminate the current grammatical and punctuation mark errors and also confirm the correct scientific English.
- The authors should write the complete terms of all abbreviations (including the instruments) before the first use in the abstract and main manuscript.
- The authors should clearly explain the innovation and importance of their work on the introduction of the manuscript. They should justify the value of the work and compare their work with previously similar published papers. They should develop the advantage and applications of this procedure. The introduction section needs to be elaborated.
- The authors should work on the scientific English of the manuscript and elaborate it.
- The authors should cite important references.

- The authors should include the brand and model of all instruments and software they used in this project.

The language of the manuscript should be elaborated. 

Author Response

July 22, 2023

Sensors

Dear editors!

          We are grateful to all the reviewers for their valuable comments and suggestions. In the revised version, all of them are considered. Below, we put reviewers' comments in bold, then responses below in regular font. In the reviewed text of the article, we have highlighted those changes. We hope that considering reviewers' comments and suggestions allowed us to improve the quality of the manuscript.

Reviewer 1:

- The authors should eliminate the current grammatical and punctuation mark errors and also confirm the correct scientific English.

- The authors should write the complete terms of all abbreviations (including the instruments) before the first use in the abstract and main manuscript.

- The authors should work on the scientific English of the manuscript and elaborate it.

Thank you for the comments!

We did our best to improve the text of the article from the point of view of grammatical mistakes and scientific English. We attached the "Certificate of English Editing". We put all abbreviations in the text and in the abbreviation section.

- The authors should clearly explain the innovation and importance of their work on the introduction of the manuscript. They should justify the value of the work and compare their work with previously similar published papers. They should develop the advantage and applications of this procedure. The introduction section needs to be elaborated.

We have substantially modified the introduction section in order to emphasize the advantages of our method and compare it with similar works. To calculate the concentrations of the components, we have used a type of OCO retrieval algorithm (see lines 253–257).

- The authors should cite important references.

The current manuscript version contains 59 references and each of them is cited.

- The authors should include the brand and model of all instruments and software they used in this project.

We have described all the components of the FTIR mockup (lines 131-179). Also, we have put the phrase "We use open source software such as Python, NumPy, SciPy, and Matplotlib for data processing and visualization" in lines 264–265.

Reviewer 2 Report

This article is about a study of the design of a 16U 11 CubeSat spacecraft equipped with a compact low-resolution FTIR spectrometer. The content of the research is interesting. At present, it seems that there is not enough data, and the logic of the article needs to be strengthened. It is recommended to reconsider acceptance after  revisions.

My comments are as follows:

1.      The introduction part does not highlight the main point of this study. The article must clearly state its research significance and novelty. The introduction should be logical and readable. Therefore, I suggest rewriting the introductory section.

2.      In Table 1, how does the author determine the Lifetime of the machine?

3.      Figure 3 should be explained in more detail.

4.      The author presents the composition of the instrument in lines 120-136. In fact, related research already exists. In this manuscript, the authors do not detail the differences between the work done and existing studies.

5.      The authors claim that the instrument will be used on spacecraft, but the study only includes data from its tests in cities. How to ensure its workability in the space environment? This should be detailed in the manuscript.

6.      In Figure 7, I suggest adding actual test pictures. And detail the experimental details in the text.

7.      It is recommended that the authors provide more valuable data to ensure the reliability of the data.

8.      The conclusion of the article should be well-organized, and a rewrite is recommended.

Moderate editing of English language required

Author Response

July 22, 2023

Sensors

Dear editors!

          We are grateful to all the reviewers for their valuable comments and suggestions. In the revised version, all of them are considered. Below, we put reviewers' comments in bold, then responses below in regular font. In the reviewed text of the article, we have highlighted those changes. We hope that considering reviewers' comments and suggestions allowed us to improve the quality of the manuscript.

Reviewer 2:

The introduction part does not highlight the main point of this study. The article must clearly state its research significance and novelty. The introduction should be logical and readable. Therefore, I suggest rewriting the introductory section.

We have substantially modified the introduction in order to emphasize the advantages of our method in comparison with similar works.

- Table 1, how does the author determine the Lifetime of the machine?

To make it clear, we have written: "The lifetime of the spacecraft is limited by the insufficient electrical power supply system parameters".

Figure 3 should be explained in more detail.

In lines 131–175, we have described Fig. 3 and the operation principle in detail.

The author presents the composition of the instrument in lines 120-136. In fact, related research already exists. In this manuscript, the authors do not detail the differences between the work done and existing studies.

In lines 45–88, we have made a comparison with the available tools and noted the novelty of our work and the motivation of the study.

The authors claim that the instrument will be used on spacecraft, but the study only includes data from its tests in cities. How to ensure its workability in the space environment? This should be detailed in the manuscript.

We have put lines 280-297 into the text as follows: "The presented results experimentally demonstrate the possibility of measuring greenhouse gas concentrations by reflected solar radiation using an IR Fourier spectrometer. The developed compact low-resolution FTIR spectrometer is designed to be placed in CubSat spacecraft for remote measurement of greenhouse gas concentrations.

The functional parts of the CubSat FTIR spectrometer are the same as for the ground-based mockup. The differences are in the mechanical design, for example, the use of special materials with an extended temperature range. To limit the radiation flux motorized diaphragm can be installed in the optical system. Diaphragm allows us to change the size of the radiation flux in the range from 1 mm to 27 mm. To ensure functioning in outer space, special heat sinks from electronic circuit boards can be used, special vacuum lubrication of moving parts is provided, and a radiation-resistant element base is used. The CubSat FTIR spectrometer will be tested in conditions close to those in outer space.

For the CubSat FTIR spectrometer, we assume the detection limit is about 1% of the nominal values of CO2 and CH4 concentrations in the atmosphere. The precision and accuracy of gas detection and concentration evaluation are determined by the spectral resolution of the device at a level of 2 cm-1. In the considered measurement range, the main atmospheric components do not have strong absorption lines according to HITRAN, GEISA databases, and SPECTRA Spectroscopy Tools".

In Figure 7, I suggest adding actual test pictures. And detail the experimental details in the text.

We have changed Fig. 7 and added the description to lines 239-252.

It is recommended that the authors provide more valuable data to ensure the reliability of the data.

Unfortunately, it is impossible to obtain additional experimental data at the moment because the spectrometer is being finalized. However, we have carried out additional theoretical assessments of sensitivity (see lines 292–297).

The conclusion of the article should be well-organized, and a rewrite is recommended.

We have substantially modified the conclusion section.

Reviewer 3 Report

I reviewed the manuscript “Determination of Greenhouse Gases Concentrations from the 16U CubeSat Spacecraft Using Fourier Transform Infrared Spectroscopy” (sensors-2476935). The idea is interesting and described correctly. I just have some points to be considered.

1) Figures 3 and 8. I suggest including definitions of each part in the Figure Captions.

2) Define concentration units, ppm is mg/m3?

3) Check punctuation Figures 2, 8 and 9. Decimal punctuation must be point instead of comma.

4) Include Analytical parameters such as: limit of detection, precision, accuracy and evaluation of some interference gases (SO2, N2, H2).

Author Response

July 22, 2023

Sensors

Dear editors!

          We are grateful to all the reviewers for their valuable comments and suggestions. In the revised version, all of them are considered. Below, we put reviewers' comments in bold, then responses below in regular font. In the reviewed text of the article, we have highlighted those changes. We hope that considering reviewers' comments and suggestions allowed us to improve the quality of the manuscript.

Reviewer 3:

  • Figures 3 and 8. I suggest including definitions of each part in the Figure Captions.

We have fixed the definitions of each part in the Figure 3 and 8 captions.

  • Define concentration units, ppm is mg/m3?

Unit ppm is parts per million (the amount of a chemical compound in relation to the substance that contains it).

Both concentration are uniquely related such as:

concentration (mg/m3) ~ concentration (ppm) x molecular weight.

In the presented manuscript, we have used concentration values in ppm or ppm•m. Also, ppm has been put into the abbreviations section.

  • Check punctuation Figures 2, 8 and 9. Decimal punctuation must be point instead of comma.

Thank you! We have corrected this mistake.

  • Include Analytical parameters such as: limit of detection, precision, accuracy and evaluation of some interference gases (SO2, N2, H2).

We have put the following text in lines 292-297: "For the CubSat FTIR spectrometer, we assume the detection limit is about 1% of the nominal values of CO2 and CH4 concentrations in the atmosphere. The precision and accuracy of gas detection and concentration evaluation are determined by the spectral resolution of the device at a level of 2 cm-1. In the considered measurement range, the main atmospheric components do not have strong absorption lines according to HITRAN, GEISA databases, and SPECTRA Spectroscopy Tools".